# Aspects of Gender and Sexuality in Relation to Experiences of Subjection to Sexual Harassment among Adolescents in General Population

**DOI:** 10.3390/ijerph19169811

**Published:** 2022-08-09

**Authors:** Riittakerttu Kaltiala

**Affiliations:** 1Faculty of Medicine and Health Technology, Tampere University, 33014 Tampere, Finland; riittakerttu.kaltiala@tuni.fi; 2Department of Adolescent Psychiatry, Tampere University Hospital, 33520 Tampere, Finland; 3Vanha Vaasa Hospital, 65380 Vaasa, Finland

**Keywords:** sexual harassment, sex differences, gender identity, sexuality

## Abstract

Subjection to sexual harassment (SH) has been reported more commonly by girls than by boys, by sexual and gender minority youth more than by mainstream youth, and by sexually active youth more than by those not yet experienced in romantic and erotic encounters. However, the research so far has not addressed these correlates simultaneously. This study aimed to explore independent associations between experiencing SH and these aspects of sex, gender and sexuality—considering all of them concurrently. A cross sectional survey with data from Finland was used, with an analyzable sample of 71,964 adolescents aged 14 to 16-years- of age, collected in 2017. The data were analysed using cross-tabulations with chi-square statistics and logistic regression analyses. The types of SH studied were gender harassment, unwelcome sexual attention, and sexual coercion. Girls, sexual and gender minority youth, and youth engaging in romantic and erotic encounters had experienced all three types of SH more commonly than boys, mainstream youth and those not sexually active. Associations between minority status and experiences of sexual harassment were stronger among boys, and being sexually active had stronger associations with subjection to sexual harassment in girls. The findings appear to support the assumption that sexual harassment serves both as a means of perpetuating heteronormativity and the sexual double standard.

## 1. Introduction

Subjection to sexual harassment is reportedly more common among girls than among boys [1,2,3,4,5], among sexually active adolescents than among those not yet engaging in romantic and erotic encounters [2,5,6], and among sexual and gender minorities than among mainstream youth [2,4,7,8]. The associations of these aspects of sex, gender and sexuality with experiences of sexual harassment have been researched in separate studies. This study aims to simultaneously explore the associations of all these aspects of sex, gender and sexuality in adolescents with experiences of sexual harassment. I first present core concepts and theoretical assumptions possibly explaining this increased risk of sexual harassment through aspects of sex, gender and sexuality, and then proceed to analyse these associations in an unprecedentedly large adolescent population dataset.

### 1.1. Definitions

Sexual harassment has been defined in law as gender-based discrimination conducive to a hostile work/school environment, potentially seriously impairing the victim’s performance at work or ability to participate in and benefit from education [9,10]. In public or mental health research, sexual harassment constitutes a form of aggressive behaviour likely to cause its victims traumatic experiences and predispose them to mental health problems [1,3,11]. Among adolescents, sexual harassment has also been studied as a developmental phenomenon: when sexual desires emerge and socializing in mixed gender groups—instead of the one-gender groups or dyads typical for late childhood—becomes more common. Aggression also assumes new forms with sexual content, particularly during the early phases of adolescence, when behavioural control and social skills are only just taking shape [12,13,14,15]. As both bullying and perpetration of sexual harassment serve purposes of social dominance, the pathway from the former to the latter during adolescence is predictable [15].

According to Fitzgerald et al. [16], sexual harassment can be divided into gender harassment, unwelcome sexual attention, and sexual coercion [16,17]. Gender harassment includes verbal and non-verbal gender–based hostile/derogatory communication or gender-related name-calling. Unwelcome sexual attention includes any sexual advances such as propositioning, touching, invitations etc. which are distasteful and unwelcome to the target and perceived as offensive. Sexual coercion includes actual sexual assault but also any behaviour attempting to extort sexual compliance by means of promises, threats or incentives [1,11,16,17]. The boundaries between sexual harassment and (child) sexual abuse may not always be clearcut. Any sexual act involving a child and an older person is deemed child sexual abuse; “older” here often refers to an age difference of three to five years so as to exclude sexual activity among peers [18]. Sexual abuse may include extreme sexual coercion such as violent rape, but also non-contact behaviours, such as exhibitionism, bordering on sexual harassment [19]. This paper uses the definition originally proposed by Fitzgerald et al. [16].

Sex will be used throughout this paper to refer to biological sex (male, female). Gender refers to the socially constructed characteristics of women, men, girls and boys generally based on the norms, behaviours and societal expectations of individuals based primarily on their sex [20]. Gender identity refers to a person’s own subjective, internal and individual experience of gender, which may or may not correspond to that person’s physiology or sex; a person’s self-perceived gender, be that male, female, or other [20,21]. The aspects of sexuality studied in this paper are sexual behaviour and sexual orientation. Sexual orientation has been defined as comprising attraction, behaviour and identification [22,23]. This paper focuses solely on attraction (see below, Measures).

### 1.2. Sex, Gender, Sexuality and the Developmental Tasks of Adolescence

According to developmental tasks theory [24,25], each developmental stage (such as infancy and early childhood, late childhood, adolescence, etc.) has its own specific developmental tasks to be accomplished in order to successfully progress to subsequent stages. Developmental tasks arise from interactions between physical development, personal attributes and societal expectations. The developmental tasks of adolescence entail accepting one’s body, adopting a gendered social role, achieving emotional independence from one’s parents, developing close relationships with peers of the same and the opposite sex, preparing for an occupation, preparing for marriage and family life, establishing a personal value or ethics system, and adopting socially responsible behaviour [24,25]. Many of these tasks relate to gender and sexuality.

Sexual development accelerates in adolescence [26]. Romantic and erotic interests and behaviour gradually mature towards adulthood [27,28,29]. The sexes differ in sexual development due to differences in pubertal timing and to actual hormonal developments [30] and also due to differences in what is socially expected of boys and girls [31]. Sexual orientation, comprising aspects of attraction, behaviour and identity [22,23], emerges and evolves during adolescence. Gender identity, an individual’s core perception of self as a male, female or between/outside the male-female continuum, is a domain of development separate from but closely linked to sexuality [32].

### 1.3. Heteronormativity and Sexual Harassment

Heteronormativity refers to the assumption that heterosexuality is the standard for defining normal sexual behaviour and that male–female differences and gender roles are the natural and immutable essentials in normal human relations [33]. Heteronormativity further assumes the primacy of heterosexuality and clearly defined norms for male and female self-expression and behaviour [34]. Heterosexism is the mechanism of perpetuating heteronormativity. Heterosexism refers to a systematic process of privileging heterosexuality over other sexualities [34,35,36]. This manifests in overt discrimination and prejudice against those not conforming to the heteronormative ideal, but also in attitudes and subtle behaviours favouring male dominance over females, heterosexuality over other sexualities, and conformation with traditional gender roles over gender-diverse identity and self-expression, for example by controlling the visibility of the various sex, gender and sexuality related groups or by condoning discriminatory expressions against or victimizing the less prioritized groups [34,37]. Regulating gender expression has been suggested to be a critical component of heteronormativity and heterosexism [34], while sexual harassment has been suggested to be a mechanism of such regulation [10]. Adolescents commonly perpetrate aggression and compete for social dominance by means of disparaging same-sex interest and gender-nonconforming self-expression [15,34,38]. Whether or not the target of such communications is assumed to belong to a sexual or gender minority, this implies negative attitudes towards gender-nonconforming appearance and behaviour and reinforces the heterosexual norm [34,38]. Even in the face of increasing equality between sexes and the improved status of sexual and gender minorities, heterosexism remains widespread [35,36]. Thus, in addition to females at large [1,3], adolescents not conforming to gender roles, including self-expression and sexual behaviour, are more likely to suffer from sexual harassment than their heterosexual gender-conforming peers [2,4,7,8]. Effeminate (i.e., in males, behaviour considered typical of a female) self-expression among boys is less readily tolerated than is masculine (qualities or appearance traditionally associated with men) self-expression among girls [39], perhaps running decidedly contrary to the heteronormative ideal. Thus, if sexual harassment among adolescents serves to maintain heteronormativity, boys belonging to gender and sexual minorities will likely experience it more than sexual and gender minority girls.

### 1.4. Transitions in Adolescence, Sexual Behaviour and Sexual Harassment

Sexual harassment among adolescents may also be a transitional, developmental phenomenon: a new form of aggression shaped attributable to intensifying romantic and erotic interests and increased socializing in mixed gender groups during the early phases of adolescence, when social skills and behavioural controls are still only taking shape [13,14,15]. Hence, those prone to aggressive behaviour extend their repertoires of aggressive communication to contents with a sexual flavour. If this is the case, it follows that adolescents sexually interested and eager to socialize in mixed-gender groups would also be most likely to become victims of peer sexual harassment, regardless of gender expression [6]. Reporting subjection to sexual harassment has been associated with the early onset of puberty and advanced pubertal maturation [40,41], early and frequent dating, romantic and erotic relationships, a greater number of partners for sex [2,5,6,9,40,42,43], and greater attractiveness and perceived personal power [41]. This seems to highlight the role of emerging sexual desires in sexually harassing behaviours and to influence who become targets of these: unwelcome attention may be attracted by appearance and behaviours signalling sexuality. If this is indeed the case, advances in dating and sexual activity rather than gender expression would be associated with experiences of sexual harassment among adolescents.

### 1.5. The Sexual Double Standard

An example of a double standard would be different groups being judged differently when engaging in similar behaviours and acts, some being socially rewarded and others penalized. Females and males are held to different standards regarding a number of behaviours, such as self-promoting behaviour, assertiveness, passivity, drinking behaviours and sexual behaviours [44,45]. The sexual double standard means that men and boys are expected to be and are socially rewarded for being sexually adventurous, whereas women and girls are expected to be sexually reticent and are socially rewarded for chastity [31,44]. Although the sexual double standard appears to diminish in cultures more advanced in equality between the sexes, it nevertheless manifests across contemporary societies [44]. In light of this sexual double standard, subjection to sexual harassment can be assumed to have stronger associations with adolescents’ own sexual activity among girls than boys.

### 1.6. The Present Study

Taken overall, experiences of subjection to sexual harassment are reportedly more common among girls [1,4], among sexually active adolescents [6], and among sexual minority and gender minority youth [4,7,8], but to the best of my knowledge, no studies have evaluated the associations between these variables and experiences of sexual harassment simultaneously in one and the same model. As aspects of sex, gender and sexuality are decidedly intertwined during adolescence, they need to be studied simultaneously to distinguish between main and confounding effects, that is, to explore if they are all independently associated with experiences of sexual harassment or if some of them level out others when studied at the same time. Therefore, this study seeks answers to the following questions:Are sex, gender identity, sexual orientation and sexual activity all independently associated with experiences of subjection to sexual harassment among adolescents in the Finnish general population?Are the associations between individuals’ self-experienced gender, sexual orientation and sexual activity and experiences of subjection to sexual harassment similar in both sexes?

## 2. Materials and Methods

The School Health Promotion Study (SHPS) is a school-based cross-sectional anonymous survey designed to examine the health, health behaviours and school experiences of teenagers across Finland. It is conducted by the National Institute for Health and Welfare. The survey items were formulated by an expert panel at the National Institute for Health and Welfare, which annually determines the content of the SHPS. The survey is sent to every municipality in Finland, and the municipalities decide if the schools in their areas will participate. Parents are informed about the survey, but according to the child ombudsman’s ruling of 2012, children are entitled to express their opinions in SHPS of their own free will [46]. The pupils are informed about the voluntary nature of participation and about their option to leave unanswered any questions or cease participating at any point. They then respond to the questionnaire online during a school lesson supervised by a teacher, who does not interfere with responding but ensures that they all have peace to respond undisturbed. A completed questionnaire is taken as consent to participate. The survey is run primarily for health policy and administrative purposes and the data is available on request for purposes of scientific research. The main aim of the survey is to produce national adolescent health indicators that municipalities can utilize in planning services and that can be used at national level to assess the effectiveness of health policies. The author obtained permission to use the data for scientific research but was not responsible for its collection. The School Health Promotion Study has received ethical approval from the Tampere University Hospital ethics committee and the ethics committee of the National Institute of Health and Welfare.

The survey is conducted among eighth and ninth graders of the comprehensive schools and among second-year students in upper secondary education during the spring term. In the present study, SHPS data from the comprehensive school sample of spring 2017 were used. Nine-year comprehensive education, starting at age seven, is compulsory in Finland. Of all the eighth and ninth graders in the country, 63% participated in the 2017 survey. The participation rate across the 19 regions of the country varied from 54% to 75% [47]. In total the participants included 73,413 adolescents: 36,195 boys, 36,815 girls, and 403 adolescents who did not report their sex. Their mean (sd) age was 14.83 (0.83) years.

### 2.1. Measures

Sexual harassment. The adolescents were asked if during the past 12 months they had experienced any of the following: (1) disturbing sexual propositions or harassment at school, during hobbies, on the street, in shopping malls or other public spaces, or via telephone or the Internet; (2) bullying, name-calling or criticism that insulted their bodies or sexuality; (3) being touched on intimate body parts against their will; (4) being pressured or coerced into intercourse or other sexual activity; and (5) being offered money, goods or drugs/alcohol in return for sexual favours, all with the response options of yes/no. The five items were classified as gender harassment, unwelcome sexual attention, and sexual coercion [16]. Gender harassment was recorded if the respondents reported that they had repeatedly experienced sexual name-calling (question 2). Unwelcome sexual attention was recorded if the respondents reported that they had repeatedly experienced disturbing sexual propositions (question 1) or being touched in intimate body parts against their will (question 3). Sexual coercion was recorded if the respondents reported that they had been repeatedly pressured or coerced into granting sexual favours or offered payment for sexual favours (questions 4 and 5) [1,11].

Sex. The respondents were first asked “What is your sex?”, with the response options of “boy” and “girl”. This was intended to elicit the respondent’s sex, as noted in their identity documents, and was the opening question of the survey. Given that sex changes are only legal after age 18 in Finland, sex as noted in identity documents reflects the person’s biological sex in this age group. “Male” and “female” are currently the two options available in official identity documents in Finland, and sex was elicited accordingly in the SHPS. Throughout this paper, sex refers to the responses to this first question. Those indicating that they were boys are referred to as boys or as males, and those indicating that they were girls are referred to as girls or as females. In the analyses, boys are used as the reference category.

Gender identity. Later, in the section of the survey addressing health, respondents were asked about their self-perceived gender as follows: “Do you perceive yourself to be …”, with response options “a boy/a girl/both/neither/my perception varies”. According to sex and self-perceived gender, the respondents were categorized into one of three gender identities: cisgender identity (indicated being male and perceiving himself as a boy, or indicated being female and perceiving herself as a girl), opposite sex identification (indicated being male, perceived identity girl; or indicated being female, perceived identity boy), and other/non-binary gender identity (independent of sex: perceived identity both boy and girl, perceived identity neither boy nor girl, variable) [48]. In the analyses, cisgender identifying adolescents are used as the reference category.

Sexual orientation was recorded according to attraction, and elicited as follows: “Have you had a crush on or been in love with …”, with response alternatives yes, girl(s)/yes, boy(s)/yes, both girl(s) and boy(s)/no I haven’t/I don’t know”. In the analyses, these were classified as attracted exclusively to the opposite sex/attracted to the same sex or both sexes/not attracted to anyone or unsure Kaltiala-Heino et al. [7]. It has previously been shown that in this age group, not having been/being unsure of whether one has been in love or had a crush on anyone can be collated to one category representing not yet being active in the domain of erotic and romantic interests, and is not a sign of nonconformity or developmental challenges [7]. As it was expected that in this group those interested in the neither/doesn’t know group would report the fewest experiences of sexual harassment [7], they are used as reference category in the analyses.

Sexual activity was measured according to reports of a steady relationship and of having experiences of sexual intercourse. Dating was elicited by the question “Are you currently in a steady relationship?” (no/yes), and having experienced intercourse by the question “Have you ever had sexual intercourse?” (no/yes). In the analyses, those not in a relationship and those with no experience of intercourse are used as reference categories. It is acknowledged that being in a steady relationship may mean different things for young people at different stages of adolescent development [27], but it is nevertheless assumed that, for an adolescent, perceiving their relationship with someone as a steady relationship likely shapes their social lives and perceptions thereof. Sexual behaviours progress during adolescence from early advances, such as holding hands, to kissing, fondling and genitally intimate behaviours. Sexual intercourse is one step in this process and may not be so relevant for sexual minority youth [49], but is nevertheless frequently used in adolescent research as an indicator of sexual debut, it being a fairly clear-cut event and entailing certain risks not associated with the earlier stages of sexuality [27,29].

### 2.2. Facetious Responding

Adolescents responding to survey studies may exaggerate their problem behaviours and negative experiences as well as their belonging to minorities, thereby creating a risk of erroneous conclusions regarding problem behaviours and negative experiences among minority youth [48]. In the Finnish school system, the timing of the survey means that eighth graders turn 15 during the year of responding, and ninth graders turn 16. Thus, the respondents are assumed to be 14 to 16 years old. It is extremely rare to repeat a class in comprehensive school, and therefore pupils are decidedly homogenous regarding age. In order to control for facetious responding, adolescents reporting that they were 13 years old or younger were excluded. Adolescents reporting that they were 17 years or older were also excluded, both in order to control for facetious responding and also to keep the studied group homogenous; if there are occasionally pupils aged 17 or more in these grades there will be special reasons for this and such young people will likely not be representative of eighth and ninth graders on average. This age rule resulted in the exclusion of 1449 respondents (2.0%). The demographic characteristics of the sample of 71,694 respondents analysed and the distribution of the study variables are presented in Table 1.

### 2.3. Statistical Analyses

Proportions of those reporting experiences of subjection to sexual harassment are presented. Differences between the groups in cross-tabulations were studied using chi-square statistics/Fisher’s exact test where appropriate. Logistic regression was used to study associations between experiences of sexual harassment and sex, gender identity, sexual orientation, having had a steady relationship and having experienced intercourse. The three forms of sexual harassment were entered each in turn as the dependent variable. Sex, gender identity, sexual orientation, having had a steady relationship and having experienced intercourse were entered as independent variables, controlling for age. Next, the analyses were run stratified for sex, obviously excluding sex from the independent variables. Odds ratios (OR) with 95% confidence intervals (CI) are reported. Due to the large size of the data, the cut-off for statistical significance was set at *p* < 0.001. Statistical significance of differences in odds ratios between boys and girls in the stratified analyses were tested using interaction analyses. The three forms of sexual harassment were each in turn entered as the dependent variable. Independent variables entered included gender identity, sexual orientation, having had a steady relationship and having experienced intercourse, controlling for age, and each in turn an interaction term sex* [gender identity/sexual orientation/steady relationship/having experienced intercourse].

## 3. Results

Of the respondents, 78.5% reported none of the sexual harassment experiences listed, while 10.6% reported experiences of gender harassment and 16.9% reported experiences of unwelcome sexual attention. Sexual coercion was reported by 4.2%. Of those reporting experiences of sexual harassment, 66.1% reported only one type, 22.7% two types and 11.3% all three types.

Proportions of those reporting experiences of gender harassment, unwelcome sexual attention and sexual coercion according to the gender and sexuality related variables are presented in Table 2.

Multivariate associations are presented in Table 3.

All three forms of sexual harassment were reported more commonly by girls than boys (ORs 1.7–3.9).

All three forms of sexual harassment were reported more commonly by adolescents reporting non-binary/other gender identity (ORs 3.0–4.7) than by cisgender identifying adolescents. Gender harassment and sexual coercion were also reported more commonly by respondents identifying with the opposite sex than by cisgender respondents (ORs 1.5–2.2).

Compared to those not attracted to anyone/unsure, those attracted to the same sex or both sexes had increased ORs (2.6–3.9) for all forms of sexual harassment, and those attracted exclusively to the opposite sex also had increased ORs for gender harassment and unwelcome sexual attention (1.4–1.6) compared to the reference group (=those not (yet) attracted to anyone).

Having experienced intercourse yielded increased ORs for all forms of sexual harassment (ORs 2.2–7.3) compared to those who had not experienced intercourse, those currently in a steady relationship for gender harassment and for unwelcome sexual attention (OR 1.2 for both) than for those not in a relationship. Age was inversely associated with reporting gender harassment. (Table 3, columns “all”).

### Stratification by Sex

The interaction terms “sex*”, “gender identity”, and “sex* sexual orientation” were statistically significantly (*p* < 0.001) associated with all three forms of sexual harassment. In addition, the interaction term “sex* having experienced intercourse” was statistically significantly (*p* < 0.001) associated with reporting experiences of sexual coercion. This means that the differences in ORs between boys and girls for all the three forms of sexual harassment according to gender identity and according to sexual orientation were statistically significant, and that the difference between boys and girls in ORs for sexual coercion according to having experienced intercourse was statistically significant.

The associations between gender identity other than cisgender and experiences of sexual harassment were stronger among male respondents (ORs 2.9–5.2 as compared to cisgender males), whereas among female respondents, even statistically significantly decreased sexual harassment experiences were noted (for unwelcome sexual attention by opposite sex identification) (ORs 0.6–1.0 compared to cisgender females).

Being attracted to the same sex or both sexes had stronger associations with experiences of sexual harassment among males (ORs 3.4–5.6 with boys not yet attracted/unsure as reference group) than females (ORs 2.5–3.7 with girls not yet attracted/unsure as reference group). Among females, both being attracted exclusively to the opposite sex and reporting same sex/both sexes attraction yielded increased odds ratios for experiences of sexual harassment compared with those not yet attracted to anyone.

The association between having experienced intercourse and reporting subjection to sexual coercion was stronger among females than males (OR 8.8 and 4.4 compared to those with no experience of intercourse, respectively; Table 3).

## 4. Discussion

Experiences of sexual harassment were reported more commonly by females, by those active in romantic and erotic encounters, and by sexual and gender minority youth. Earlier research has noted the associations between experiences of subjection to sexual harassment and sex [1,3], sexual activity [2,5,6], and sexual or gender minority status [2,4,7,8]. The novel contribution of the present study is that, as far as I know, the associations between these sex, gender and sexuality-related variables and experiences of sexual harassment were all explored here for the first time concurrently, which makes it possible to ascertain if they all have independent associations when the others are also accounted for. The gender and sexuality-related variables had somewhat different associations with experiences of sexual harassment among boys and girls. I next compare these findings to those of earlier research in the field and discuss their implications.

Sexual harassment experiences were unevenly distributed across sex, gender identity and sexual orientation. This lends support to heterosexism as a motivation for sexual harassment among adolescents. Consistent with supporting a dominant position of males over females, heterosexuality over other sexualities, and cisgender identity over gender minorities, sexual harassment appeared to be targeted more commonly at females than males, and also more at non-heterosexual and non-cisgender adolescents than at heterosexual and cisgender youth. Sexual harassment has long been recognized as a form of victimization of women by men, attributed to and strengthened by the notion of male superiority [10]. The concepts of heteronormativity and heterosexism extend the understanding of inequality between sexes to inequality between sexualities and gender expressions. Self-expression not conforming to binary sex roles may trigger aggression that serves to maintain heteronormativity [8,50]. In the present study, experiences of subjection to sexual harassment had the strongest associations with non-binary/other gender identity, which is likely accompanied by the most marked challenge to gender-conforming self-expression. Similarly, in excessively targeting adolescents interested in the same sex or both sexes, sexual harassment appears to be a mechanism of penalizing those who do not comply with the heteronormative order, and meets the definition of heterosexist discrimination, which refers to harassment, rejection, and unfair treatment due to one’s sexual orientation [35].

A statistically significant, although weak, association was also detected between currently going steady and the subjection to sexual harassment. Steady relationships among early and middle adolescents occur predominantly in the context of socializing in mixed-gender groups, and thus sexual harassment among adolescents may in part represent inept expressions of attraction during a developmental phase when behavioural control and social skills are immature [13,14,15,27]. Subjection to all forms of sexual harassment was least commonly reported by those adolescents who reported romantic interest in neither girls nor boys or by those who did not know if they had been in love or had a crush. Becoming interested and involved in romantic encounters may both bring adolescents into contact with peer groups where sexually harassing behaviours occur and also make them more perceptive to sexual cues and hence more likely to pay attention to, recall and report experiences of sexual harassment [6]. Likewise, subjection to sexual harassment was more commonly reported by those who had experienced intercourse.

Among girls, mainstream romantic attraction to the opposite sex already increased ORs for reporting subjection to sexual harassment and the interaction analyses revealed that the association between being sexually active (having experienced intercourse) and subjection to sexual coercion was much stronger among girls than among boys. With these findings, the present study also lends support to the existence of a sexual double standard, resulting in males being socially rewarded for sexual activity and females for chastity [31,44]. Sexual harassment targeting particularly sexually active females suggests that such harassment may serve as an attempt to control female sexuality. Finally, the sexual double standard may in itself be seen as arising from heteronormativity and as reinforcing clearly defined and separate male and female roles, as well as male dominance [44].

The interaction analyses also revealed that belonging to a sexual or gender minority had stronger associations with experiences of sexual harassment among males than among females. This further supports the theory that sexual harassment serves as a means of reinforcing heteronormativity. Opposite sex identification and same-sex romantic and erotic interest may both be associated with unconventional behaviour in relation to sex roles [51]. Boys not conforming to heteronormative expectations of how boys should behave may be perceived as a threat to male superiority and precipitate attempts to reinstate compliance to the behavioural norms, for example, by sexual harassment. Masculine behaviour in girls is better tolerated than effeminate self-expression in boys [39]. Indeed, opposite sex identification among girls was even associated with decreased experiences of sexual harassment compared to cisgender identity.

The findings have a number of implications. Inequality is a serious social problem. Tackling the mechanisms that perpetuate inequality is imperative. In addition to sexual harassment being a means of reinforcing inequality, subjection to it may be a traumatic event constituting a risk to adolescent development. Sexual harassment is associated with a number of psychosocial and mental health problems in adolescence [3,11,52,53], and appears to be as harmful or even more so to adolescents than other forms of harassment [3,52]. This further underlines the need to eradicate sexual harassment in the interests of adolescents’ mental health and well-being. Interventions are needed on many levels. At the societal level, policies and choices conducive to equality between sexes, sexualities and identities in general, and universal sexual rights [54] in particular, need to be promoted. School is an important environment in which to work to reduce sexual harassment among adolescents. For example, teachers and other school personnel do not always recognize the heterosexist language commonly used by adolescents [55]. Heightened awareness should of course be followed by appropriate action. The school personnel need to be ready to intervene constructively [56], and studies have shown that a school climate including a clear disciplinary structure (strict but fair rules) and high student support (students feel supported and respected by school staff) is associated with lower levels of peer-to-peer sexual harassment among pupils [57]. School belonging again buffers against the negative impact of sexual harassment [58]. Deconstructing internalized heterosexism, reducing negative (such as depressive or anger rumination) and enhancing positive (such as self-compassion) coping responses may be a relevant focus for counselling [59,60] in order to empower young people to cope effectively and resist a sexually harassing culture and eradicate trauma related to sexual harassment on an individual level.

### Methodological Considerations

The uniquely large population sample is a major strength of the present study. The coverage of compulsory comprehensive education in Finland is almost 99%. The study did not reach adolescents who were not at school on the survey day. Such absences are known to account for 10–15% of those enrolled in schools. Among those absent, psychosocial problems of all kinds may be more common than among attending pupils [61]. However, not even major attrition need necessarily bias conclusions on the relationships between the phenomena studied in survey studies [62].

Variation in participation rates between regions is a result of the municipalities’ decisions whether or not to participate in the SHPS, and as such is not likely to bias findings on the phenomena reported on in this paper.

Adolescent sexual behaviours comprise much more than intercourse and advance stepwise over the adolescent years [27]. Experiences of sexual harassment also increase stepwise as sexual behaviours advance [6]. However, intercourse is often used in adolescent studies as an indicator of sexual debut. It is also an important milestone given the particular requirements for maturity related to it and also the risks, such as unwanted pregnancy and sexually transmitted diseases [28]. Intercourse may be considered a weaker indicator of sexual development among sexual minority youth [49]. However, research has shown that adolescents identifying with sexual minorities may have experiences of intercourse even more commonly and earlier than heterosexually identifying peers [63,64].

Sexual orientation has been defined as comprising aspects of attraction, behaviour and identification [22,23]. This study focused solely on attraction. A more comprehensive understanding of sexual orientation might have resulted if all aspects of sexual orientation had been elicited. However, given that self-identification as LGB rather follows than precedes romantic and erotic same-sex attractions [22], eliciting romantic and/or sexual attractions is developmentally more appropriate in the age groups studied than asking about sexual identity. A limitation may be that attraction was only elicited as to the sex of the object of such attraction, not their gender identity.

As has been recommended [65,66], a two-step approach to identify subjects with differing gender identities was taken, eliciting sex and gender self-perception separately. Sex was elicited at the very beginning of the survey, and experienced gender later, in the section on health. As regards the first question of the survey, the possibility that some respondents identifying strongly with the opposite sex actually reported self-perceived gender rather than sex, as noted in identity documents, cannot be excluded. This methodological concern has also been acknowledged in earlier corresponding studies [65,67]. This is a limitation inherent in the anonymous survey method and a limitation in the present study.

The present data only permits defining categorical variables for sexual orientation (interested solely in opposite sex/solely in same sex/both/none) and gender identity (cisgender/opposite sex identification/non-binary or other). More detailed analyses of the associations between gender experience, sexual orientation and victimization would be possible if non-binary or dimensional variables were available. A non-binary approach is recommended for future studies.

One strength of the present study is controlling for implausible and potentially facetious responding by excluding adolescents reporting ages too young or too old for the school grades in question. It has been shown that spurious responding particularly influences topics related to minority status and negative experiences, but also to sexuality in general [6,7,48,68], and indeed in the present study, those responses excluded due to reporting improbable age had more commonly reported being gender and sexual minorities, being sexually active and being in steady dating relationships. It is important to use strategies to exclude facetious responding in order to avoid erroneously negative conclusions about minorities.

A limitation of the present study is the cross-sectional design, which allows for no conclusions on causal relationships. I have discussed the possibilities of how the associations detected may be created, but future studies with longitudinal designs are needed to be able to determine causality.

## 5. Conclusions

The findings of the present study support the assumption that sexual harassment may serve as a means to sustain heteronormativity and the sexual double standard. Being female, of a sexual or gender minority and romantically and sexually active were all independently associated in this study, with the increased likelihood of experiencing sexual harassment, but the associations were somewhat different among boys and girls. Deconstructing the sexual double standard and heteronormativity with its inherent subordinations is called for.

## Figures and Tables

**Table 1 ijerph-19-09811-t001:** Distributions of sex, gender and sexuality related variables and sociodemographic backgrounds and mean age in years among eighth and ninth graders of comprehensive schools participating in the School Health Promotion Survey in Finland in 2017.

		**N**	**(%)**
Sex			
	Male	35,449	(49.3)
	Female	36,146	(50.2)
	Missing	369	(0.5)
Gender identity			
	Cisgender	67,105	(93.2)
	Opposite sex	562	(0.8)
	Non-binary/other	3240	(4.5)
	Missing	1057	(1.5)
Attracted to			
	Exclusively opposite sex	49,942	(69.4)
	Same sex or both	4497	(6.2)
	None/does not know	16,179	(22.5)
	Missing	1346	(1.9)
Currently in steady relationship			
	No	58,684	(81.5)
	Yes	11,993	(16.7)
	Missing	1287	(1.8)
Ever had sexual intercourse			
	No	56,662	(78.7)
	Yes	12,857	(17.9)
	Missing	2445	(3.4)
		M	(SD)
Age, years		14.84	(0.70)
		**N**	**(%)**
Living with both parents			
	Yes	47,709	(66.3)
	No	20,984	(29.2)
	Missing	3271	(4.5)
Mother has only basic education			
	No	60,349	(83.9)
	Yes	4009	(5.6)
	Missing	7606	(10.6)
Father has only basic education			
	no	57,432	(79.8)
	Yes	5712	(7.9)
	Missing	8820	(12.3)
At least one parent laid off or unemployed past 12 months			
	No	46,875	(65.1)
	Yes	21,052	(29.3)
	Missing	4037	(5.6)

**Table 2 ijerph-19-09811-t002:** Proportions of adolescents reporting experiences of gender harassment, unwelcome sexual attention and sexual coercion according to gender and sexuality related variables (% (n/N)).

		Gender Harassment ______________________	Unwelcome Attention ______________________	Sexual Coercion ______________________
Sex				
	male female	7.9 (2622/33,130) 13.2 (4643/35,279) *p* < 0.001	8.5 (3022/35,449) 25.1 (9807/36,146) *p* < 0.001	3.4 (1212/35,449) 4.9 (1785/36,146) *p* < 0.001
Gender identity				
	cisgender opposite sex non-binary/other	9.4 (6099/64,699) 26.3 (134/509) 34.8 (1013/2907) *p* < 0.001	15.8 (10,577/67,105) 28.3 (159/562) 41.0 (1327/3240) *p* < 0.001	3.3 (2216/67,105) 14.1 (79/562) 21.0 (680/3240) *p* < 0.001
Attracted to				
	none/does not know opposite sex same sex/both sexes	6.4 (993/15,598) 10.0 (4807/48,210) 33.6 (1433/4268) *p* < 0.001	9.3 (1510/16,179) 17.4 (8705/49,942) 40.7 (1830/4497) *p* < 0.001	2.1 (345/16,179) 3.8 (1916/49,942) 15.8 (709/4497) *p* < 0.001
Currently in steady relationship				
	no yes	9.2 (5213/56,701) 17.7. (2808/11,431) *p* < 0.001	14.7 (8598/58,684) 28.8 (3459/11,993) *p* < 0.001	3.0 (1785/58,684) 9.9 (1187/1193) *p* < 0.001
Ever had intercourse				
	no yes	8.6 (4742/54,973) 19.9 (2403/12,098) *p* < 0.001	13.6 (7687/56,662) 32.4 (4166/12,857) *p* < 0.001	2.0 (1135/56,662) 14.0 (1799/12,857) *p* < 0.001

**Table 3 ijerph-19-09811-t003:** Multivariate associations between gender and sexuality related variables and experiences of subjection to three forms of sexual harassment among 14–16-year-old Finnish adolescents.

		Gender Harassment			Unwelcome Sexual Attention			Sexual Coercion		
		All	Males	Females	All	Males	Females	All	Males	Females
		OR (95%CI)	OR (95%CI)	OR (95%CI)	OR (95%CI)	OR (95%CI)	OR (95%CI)	OR (95%CI)	OR (95%CI)	OR (95%CI)
Sex										
	Male	ref			ref			ref		
	Female	**1.7 (1.6–1.8)**			**3.9 (3.7–4.1)**			**1.7 (1.6–1.9)**		
Gender identity										
	Cisgender	ref	ref *	ref *	ref	ref *	ref *	ref	ref *	ref *
	Opposite sex	**1.5 (1.2–1.9)**	**2.9 (2.0–4.1)**	1.0 (0.7–1.3)	1.2 (0.9–1.5)	**3.8 (2.8–5.3)**	**0.6 (0.4–0.7)**	**2.2 (1.6–2.9)**	**5.2 (3.5–7.6)**	0.9 (0.6–1.4)
	Non-binary/other	**3.9 (3.6–4.4)**	**4.2 (3.7–4.9)**	**2.2 (1.9–2.5)**	**3.0 (2.7–3.2)**	**5.7 (5.0–6.6)**	**1.5 (1.3–1.7)**	**4.7 (4.1–5.3)**	**8.2 (6.9–9.7)**	**2.3 (1.9–2.8)**
Attracted to										
	None/does not know	ref	ref *	ref *	ref	ref *	ref *	ref	ref *	ref *
	Opposite sex	**1.4 (1.3–1.5)**	1.2 (1.1–1.4)	**1.5 (1.4–1.7)**	**1.6 (1.5–1.8)**	1.2 (1.1–1.3)	**1.9 (1.7–2.0)**	1.1(1.0–1.3)	0.8 (0.7–1.0)	**1.5 (1.2–1.8)**
	Same sex/both sexes	**3.9 (3.6–3.4)**	**5.6 (4.7–6.7)**	**3.7 (3.2–4.2)**	**3.3 (3.0–3.6)**	**4.3 (3.6–5.1)**	**3.2 (2.9–3.5)**	**2.6 (2.2–3.0)**	**3.4 (2.7–4.3)**	**2.5 (1.9–3.1)**
Currently in steady relationship										
	No	ref	ref	ref	ref	ref	ref	ref	ref	ref
	Yes	**1.2 (1.1–1.3)**	1.1 (1.0–1.2)	**1.2 (1.1–1.3)**	**1.2 (1.1–1.2)**	1.2 (1.0–1.3)	**1.1 (1.1–1.2)**	1.0 (1.0–1.1)	**1.4 (1.2–1.6)**	0.8 (0.7–0.9)
Ever had intercourse										
	No	ref	ref	ref	ref	ref	ref	ref	ref *	ref *
	Yes	**2.2 (2.1–2.4)**	**1.9 (1.7–2.1)**	**2.3 (2.1–2.5)**	**2.7 (2.6–2.9)**	**2.5 (2.2–2.7)**	**2.7 (2.5–2.9)**	**7.3 (6.6–8.0)**	**4.4 (3.8–5.2)**	**8.8 (7.8–10.0)**
Age		**0.9 (0.9–0.9**)	**0.9 (0.8–0.9)**	**0.9 (0.9–1.0)**	1.0 (1.0–1.1)	1.0 (0.9–1.0)	1.1 (1.0.–1.1)	0.9 (0.9–1.0)	0.9 (0.8–1.0)	0.9 (0.9–1.0)

Note. Odds ratios (95% confidence intervals) statistically significant at level *p* < 0.001 are highlighted in bold. * Based on interaction analyses, odds ratios differ statistically significantly between boys and girls.

## Data Availability

Researchers can request data from Findata https://findata.fi/.

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
