# Peer review of "Aspects of Gender and Sexuality in Relation to Experiences of Subjection to Sexual Harassment among Adolescents in General Population"

_ijerph, 2022, doi:10.3390/ijerph19169811_

Round 1

Reviewer 1 Report

The paper has got a big sample, a clear method and nice results about a very interesting topic. Anyway, I would like to make some comments trying to improve the quality of the paper:

- The abstract is written in a very confusing way, it would be interesting to write it again more clearly and trying not to repeat the exact words continuously. 

- I highly recommend using other references to justify the topic than the authors'' previous papers

- Ther is a huge limitation and a lost chance in the paper: the use of binary categories with a sample with national representativity decrease the quality of the data, because with this social representation sample it could be possible to indicate the identification with traditional roles & alternative roles and the connection with other factors such as violence

- There are references very confusing for a scientific paper, such as Psychology Today (line 80), Adelson & American Academy of Child & Adolescent Psychiatry Committee on Quality Issues, 2012 (line 83)...

Author Response

The paper has got a big sample, a clear method and nice results about a very interesting topic. Anyway, I would like to make some comments trying to improve the quality of the paper:

- The abstract is written in a very confusing way, it would be interesting to write it again more clearly and trying not to repeat the exact words continuously. 

RE: I tried to follow the style of a structured abstract but without headings. I added information about the data and attempted to reduce repeating exact words.

- I highly recommend using other references to justify the topic than the authors'' previous papers

RE: Other references were added.

- Ther is a huge limitation and a lost chance in the paper: the use of binary categories with a sample with national representativity decrease the quality of the data, because with this social representation sample it could be possible to indicate the identification with traditional roles & alternative roles and the connection with other factors such as violence

RE: Yest, such limitation is present, and unfortunately the data cannot be changed. Almost all variables in the School Health Promotion Survey are categorical and do not allow analysing finer nuances.

I am not quite sure if you also suggest widening the scope of the present article (to various forms of violence) or just notice that were the variables not binary, there would be ample opportunities?

I added a comment on this shortcoming in Methodological considerations.

- There are references very confusing for a scientific paper, such as Psychology Today (line 80), Adelson & American Academy of Child & Adolescent Psychiatry Committee on Quality Issues, 2012 (line 83)...

RE: Psychology Today refers to psychological dictionary definition of a term;  I did my best to comply with instructions on how to refer to a website. Text reference Adelson & American… was incorrectly given, now corrected.

Reviewer 2 Report

Thanks for giving me the chance to review this paper, and this is an interesting paper that addresses an important topic. I enjoy reading it, and I would like to share with the author(s) here my comments.

First, the introduction part is a bit loose with just compilation of concepts rather than talk about existent studies. It would be better if the author(s) could talk about current literature after introducing the concepts. Also, it would be much clearer if the author could align every concept together and give us a whole story. 

Second, the literature in the discussion part is better within 5 years, so that we know the latest trend of studies in this area.

Third, the conclusion part mentions about heteronormativity, which is a vital concept that underpins this research, therefore, I suggest the author(s) to add more literature regarding heteronormativity and also go deeper in the discussion part to talk about how the results supports heteronormativity.

Author Response

Thanks for giving me the chance to review this paper, and this is an interesting paper that addresses an important topic. I enjoy reading it, and I would like to share with the author(s) here my comments.

First, the introduction part is a bit loose with just compilation of concepts rather than talk about existent studies. It would be better if the author(s) could talk about current literature after introducing the concepts. Also, it would be much clearer if the author could align every concept together and give us a whole story. 

RE: I have attempted to do just this, and may be it can be accepted as another reviewer was very satisfied with the Introduction? If not, I would really appreciate a more detailed instruction of how to do this. 

Second, the literature in the discussion part is better within 5 years, so that we know the latest trend of studies in this area.

RE: More recent references were added.

Third, the conclusion part mentions about heteronormativity, which is a vital concept that underpins this research, therefore, I suggest the author(s) to add more literature regarding heteronormativity and also go deeper in the discussion part to talk about how the results supports heteronormativity.

RE: I increased references in Introduction and clarified more in the second paragraph of Discussion.

Reviewer 3 Report

AUTHORS

The paper deserves attention for many reasons. Mention should be made of the originality of the topic, the exceptionally careful theoretical introduction, the timeliness of the data and, above all, the unprecedentedly large sample (73,000). The collected material also provides information on gender identity and other sexuality-related variables. However, in many places there should be corrections, including editorial ones.

Although editorial issues are pointed out at the end of the review, here the failure to adapt the text to the requirements of the journal is noticeable. References are cited by authors' names and the list of references is arranged in alphabetical order. I strongly urge to read the instructions for authors, as well as to become more familiar with the technique of inserting text and tables into the MDPI template. The authors define three outcome indicators: gender harassment, unwelcome attention, sexual coercion. These types of experiences are non-exclusive. Can you describe their co-occurrence and provide the overall percentage of having/not having negative experiences. A chart by gender could also be added.

The abstract needs to be completely rewritten. Specific information should be included, such as the country and year of the study, the sample size, the level of harassment rates obtained and their main predictors.

The presentation of the interaction effect is rather confusing. Please clarify or change it. There can be a main effect and a 2-way interaction effect. There is no data regarding interaction, only asterisk labelling that it was studied. Table 3 should include main effects. I suggest that the table with interactions be included in the appendix as additional electronic material. Or add interaction data in the main text.

Minor strictly editorial comments

In Table 2, it is worth adding a total row. I have the impression that some information from the headline was cut off.

In Table 3, it is worth moving the confidence intervals completely to the next row next to the appropriate variable.

Line 175 - question mark omitted.

Table 1 - line "female". Please check the value 89807. One digit too many, I guess

Table 1 – line attracted to, no "/" sign in the column corresponding to gender harassment

Table - in steady relationship, please delete additional dot in the column corresponding to gender harassment.

Generally, in tables, horizontal lines should be limited and more attention paid to table breaks between pages (on page 7, duplicated headline of table 1 in the middle of the printout).

Author Response

The paper deserves attention for many reasons. Mention should be made of the originality of the topic, the exceptionally careful theoretical introduction, the timeliness of the data and, above all, the unprecedentedly large sample (73,000). The collected material also provides information on gender identity and other sexuality-related variables. However, in many places there should be corrections, including editorial ones.

Although editorial issues are pointed out at the end of the review, here the failure to adapt the text to the requirements of the journal is noticeable. References are cited by authors' names and the list of references is arranged in alphabetical order. I strongly urge to read the instructions for authors, as well as to become more familiar with the technique of inserting text and tables into the MDPI template.

RE: This was done as well as I best was able. I had got the impression that in the first submission phase, the manuscript need not yet be totally in the correct format, for example reference style would be free in initial phase.

The authors define three outcome indicators: gender harassment, unwelcome attention, sexual coercion. These types of experiences are non-exclusive. Can you describe their co-occurrence and provide the overall percentage of having/not having negative experiences. A chart by gender could also be added.

RE: I added information about percentage reporting no SH experiences and about reporting one, two or three types of it in the beginning of Results section.

The abstract needs to be completely rewritten. Specific information should be included, such as the country and year of the study, the sample size, the level of harassment rates obtained and their main predictors.

Re: I have done my best to include more precise information, yet without exceeding the word limit for abstract.

The presentation of the interaction effect is rather confusing. Please clarify or change it. There can be a main effect and a 2-way interaction effect. There is no data regarding interaction, only asterisk labelling that it was studied. Table 3 should include main effects. I suggest that the table with interactions be included in the manuscript

RE: I wrote the description of the interaction analysis in more detail in Statical analysis. The ORs in Table 3 are main effects of the study variables from the logistic regression models without any interaction term. I did my best to express the results of the interaction analyses in the text in Results section, and suggest that if you don’t mind I do not additionally use space for an extra table, as clarifying this issues takes relatively little space in text.

Minor strictly editorial comments

In Table 2, it is worth adding a total row. I have the impression that some information from the headline was cut off.

RE: The Table had changed when placed in vertically positioned paged. I placed it in the end of the manuscript on a horizontally positioned page, which made it much more pleasant. It appears complete to me as it is now.

In Table 3, it is worth moving the confidence intervals completely to the next row next to the appropriate variable.

RE: Please see above. I placed the Table on a horizontally positioned page, and now the CI:s are neatly wholly on the same row as the OR:s.

Line 175 - question mark omitted.

RE: added.

Table 1 - line "female". Please check the value 89807. One digit too many, I guess

RE: Yes, an extra digit. I am sorry for inconvenience. Now corrected.

Table 1 – line attracted to, no "/" sign in the column corresponding to gender harassment

RE: The comment must actually have concerned table 2 or 3, and appears to me to have disappeared by placing the Tables on horizontal pages.

Table - in steady relationship, please delete additional dot in the column corresponding to gender harassment.

RE: Please see above.

Generally, in tables, horizontal lines should be limited and more attention paid to table breaks between pages (on page 7, duplicated headline of table 1 in the middle of the printout).

RE: I am afraid this problem was created when the manuscript was placed in the appropriate template, but as I replaced the Tables in the end of the  manuscript, on horizontal pages when appropriate, the problem as disappeared.

Round 2

Reviewer 2 Report

I think the authors have done a good job of revision, and I recommend this paper to be accepted.